# Exercise, mTOR Activation, and Potential Impacts on the Liver in Rodents

**DOI:** 10.3390/biology13060362

**Published:** 2024-05-22

**Authors:** Giuliano Moreto Onaka, Marianna Rabelo de Carvalho, Patricia Kubalaki Onaka, Claudiane Maria Barbosa, Paula Felippe Martinez, Silvio Assis de Oliveira-Junior

**Affiliations:** 1Graduate Program in Health and Development in the Midwest Region, Federal University of Mato Grosso do Sul—UFMS, Campo Grande 79070-900, MS, Brazil; gmonaka@hotmail.com (G.M.O.); paula.martinez@ufms.br (P.F.M.); 2Graduate Program in Education and Health, State University of Mato Grosso do Sul, Dourados 79804-970, MS, Brazil; 3Graduate Program in Movement Sciences, Federal University of Mato Grosso do Sul—UFMS, Campo Grande 79070-900, MS, Brazil; claudiane.barbosa@ufms.br

**Keywords:** mTOR, exercise, liver, metabolism, fatty liver, cell signaling

## Abstract

**Simple Summary:**

Despite the consensus regarding the association between exercise training and adaptive alterations, the potential biological effects on the liver have yet to be elucidated. In particular, the modulation of the mechanistic target of rapamycin (mTOR) is known to be involved in controlling the processes of glucose and lipid homeostasis, as well as hepatic morphological changes. This narrative review aimed to describe how exercise training interventions affect signaling pathways related to mTOR, its modulation, and other effects on the liver tissue in experimental models.

**Abstract:**

The literature offers a consensus on the association between exercise training (ET) protocols based on the adequate parameters of intensity and frequency, and several adaptive alterations in the liver. Indeed, regular ET can reverse glucose and lipid metabolism disorders, especially from aerobic modalities, which can decrease intrahepatic fat formation. In terms of molecular mechanisms, the regulation of hepatic fat formation would be directly related to the modulation of the mechanistic target of rapamycin (mTOR), which would be stimulated by insulin signaling and Akt activation, from the following three different primary signaling pathways: (I) growth factor, (II) energy/ATP-sensitive, and (III) amino acid-sensitive signaling pathways, respectively. Hyperactivation of the Akt/mTORC1 pathway induces lipogenesis by regulating the action of sterol regulatory element binding protein-1 (SREBP-1). Exercise training interventions have been associated with multiple metabolic and tissue benefits. However, it is worth highlighting that the mTOR signaling in the liver in response to exercise interventions remains unclear. Hepatic adaptive alterations seem to be most outstanding when sustained by chronic interventions or high-intensity exercise protocols.

## 1. Introduction

Classically, the literature agrees that exercise training interventions supported by adequate parameters of intensity and frequency are associated with several health benefits [1]. Evidence shows that early introduction of ET can revert some metabolic disorders, such as hypercholesterolemia, insulin resistance, and fat deposition not only in humans [2,3] but also in experimental models [4,5]. Moreover, exercise training effects are mediated by adaptive changes not only in skeletal muscles but also in multiple body tissues, including the heart, adipose tissue, nervous system, and liver [1]. Regarding the liver, exercise training attenuates obesity-induced insulin resistance by the inhibitory regulation of the cardiolipin synthase 1 (CRLS1)/interferon-regulatory factor-2 binding protein 2 (IRF2bp2)-activating transcription factor 3 (ATF3) pathway cascade [6]. Resistance exercise training results in higher protein expression of sqstm1/p62 (sequestosome 1), a peptide involved in the autophagy process within the liver [7]. In addition, exercise results in greater mRNA expression of peroxisome proliferator-activated receptor-gamma coactivator-1 (PGC-1), which is an inducible co-regulator of nuclear receptors involved in a wide variety of biological responses, such as glucose and lipid metabolism, as well as in reducing triacylglycerol levels in the liver and plasmatic glycemia in diet-induced obese rats [8]. Similarly, other authors have observed a reduction in hepatic triacylglycerol levels accompanied by insulin metabolism improvements in response to a 4-week voluntary running intervention in an experimental model submitted to a high-fat diet [9]. Furthermore, based on its reducing effect on intrahepatic fat deposition, exercise training interventions are commonly recommended to prevent non-alcoholic hepatic steatosis in rodents [10], as well as the progression of non-alcoholic fatty liver disease (NAFLD), restoring liver function in multiple experimental models [10,11,12,13,14,15,16,17].

In turn, the mechanisms involved in regulating intrahepatic lipid stores in response to exercise training protocols are not fully understood. It is postulated that myokines secreted in skeletal muscles can cause multiple paracrine effects [18,19], including the regulation of energy metabolism in the liver [20]. Indeed, the regulation of hepatic fat formation is directly related to the modulation of the mechanistic target of rapamycin, better known as mTOR [1,21]. Thus, this narrative review aims to describe how exercise training interventions affect signaling pathways related to mTOR, its modulation, and other effects on the liver tissue in experimental models.

## 2. mTOR Protein (Mammalian Target of Rapamycin) or Mechanistic Targeting of Rapamycin

The mTOR protein was discovered in the early 1990s as a target for the antifungal drug rapamycin [22,23,24]. It is a serine/threonine-activated protein kinase belonging to the kinase family related to the phosphoinositide 3-kinase protein (PI3K), and it is found in multiple body tissues [25,26]. The mTOR can be stimulated in response to energy changes [27], nutrients [28], and growth factors [29]. Multiple conditions related to mTOR signaling processes demonstrate its physiological importance, including cancer [30,31], obesity [32], diabetes [33], and aging [34]. 

Many studies have described the mTOR protein structure and its actions. In general, the mTOR protein refers to the catalytic subunit of two distinct complexes, known as mTOR complex 1 (mTORC1) and mTOR complex 2 (mTORC2) [35,36], well detailed in review articles [25,26,37,38]. Both mTOR complexes have a large molecular structure. The mTORC1 complex comprises the following six known protein components: regulatory-associated protein of mammalian target of rapamycin (Raptor) [35], proline-rich Akt substrate of 40 kDa (PRAS40) [39], serine/threonine kinase mTOR [40], DEP domain-containing mTOR-interacting protein (DEPTOR) [41], mammalian lethal with SEC13 protein 8 (mLST8) [42], and tti1/tel2 [43]. In contrast, the mTORC2 molecule encompasses the following seven known protein components: serine/threonine kinase mTOR [40], rapamycin-insensitive companion of mTOR (Rictor) [44], mammalian stress-activated map kinase-interacting protein 1 (mSin1) [45], protein observed with Rictor 1 and 2 (protor1/2) [46], DEPTOR [41], mLST8 [42], and tti1/tel2 [43]. 

Instead of Raptor, Rictor (rapamycin-insensitive companion of mTOR) is one of the core components of the mTORC2, in addition to mTOR and mLST8 [44,47]. Basically, the Raptor acts in recruitment and correct subcellular localization of the substrate to mTORC1, whereas PRAS40 performs a negative regulation of this complex, according to a previous review [48]. In addition, DEPTOR is a negative mTOR regulator, inhibiting hepatic lipogenesis [49]. Generally, DEPTOR downregulation results in increased mTOR kinase activity, leading to higher levels of the phosphorylated forms of both mTORC1 and mTORC2, as well as S6K1 (Ribosomal protein S6 kinase beta-1) and Akt [41] (Figure 1). Additional and more complete information in terms of structure and functional details are better described in previous studies [23,24,25,37,38,43,47,48,50,51]. 

Indeed, mTORC1 is sensitive to rapamycin and its effects include protein and lipid synthesis, as well as the inhibition of autophagy and lysosome biogenesis [52,53] in response to changes in growth factors, amino acids, stress, oxygen levels, and energy status. These effects partly occur from the molecular signaling cascades that involve the stimulation of PI3K (phosphatidylinositol 3-kinase) followed by Akt (protein kinase B) and Ras–MAPK (a monomeric subtype of the mitogen-activated protein kinase—MAPK). Afterward, Akt and RSK (ribosomal protein S6 kinase) peptides are phosphorylated, respectively, from the PI3K/Akt and Ras–MAPK pathways, as observed in cancer conditions [54]. To sustain cell proliferation through these pathways, adequate nutrients, energy, and macromolecule stimuli are required to support a high demand for cell replication. In this context, the mTORC1 peptide is highly responsive to intracellular ATP, glucose, and certain amino acids, such as leucine, arginine, and glutamine [36,54]. Importantly, these processes are detailed in previous studies [25,55].

On that basis, mTORC1 is an important regulator of lipid metabolism in the liver for promoting lipogenesis and inhibiting lipophagy [56]. Exacerbated mTORC1 activation may impair hepatic metabolism, facilitating lipid accumulation in the hepatocytes, which has a strong correlation with the incidence of obesity, diabetes, and fatty liver disease [56]. Also, selective inhibition of mTORC1 signaling might protect against non-alcoholic fatty liver disease (NAFLD) and non-alcoholic steatohepatitis (NASH) [56]. Comparatively, much less is known about mTORC2 concerning the mTORC1 complex. More recent evidence has shown that mTORC2 is involved with multiple molecular processes, such as stimulating cell growth [57] and survival [58], metabolism [59], and cytoskeleton remodeling [60].

In general, the mTORC2 complex is less sensitive to acute treatment with rapamycin [44,47]. In contrast, chronic exposure to rapamycin is associated with changes in the mTORC2 structure, resulting in lower signaling in some cell types [61,62]. Furthermore, its signaling is not sensitive to nutrients but responsive to growth factors, such as insulin, which is accompanied by the PI3K activation mechanisms [43,45,51,63,64]. Within a phenotypical context, mTORC2 activation improves acute and chronic liver failure by inhibiting apoptosis [65], besides promoting protective effects against liver fibrosis [66]. Furthermore, some evidence has pointed out several benefits from mTORC2 inhibition, including protection against obesity and metabolic diseases in response to changes in brown adipose tissue metabolism, from lipogenic to more oxidative [67]. Furthermore, hepatic mTORC2 deficiency has sustained insulin resistance as its primary effect [68]. 

## 3. mTOR Signaling Cascade and Potential Effects on the Liver

Studies have extensively addressed the expression and activation of mTOR peptides and their adaptive effects on skeletal [69,70] and cardiac striated musculature [71,72]. However, signaling pathways and potential effects in the liver have yet to be properly studied and clarified. Usually, mTOR signaling regulates glucose and lipid homeostasis in response to fasting and feeding conditions. As previously documented [73,74,75], at least the three following primary signaling pathways are known to modulate the mTOR actions: (I) the growth factor pathway, (II) the energy pathway, and (III) the amino acid sensor (Figure 2).

During postprandial states, the mTORC1 protein is involved in anabolic processes such as hepatic lipogenesis [76,77,78]. Hyperinsulinemia induces mTOR activation due to feeding, insulin signaling, and consequent Akt stimulation [79]. Insulin-linked signaling cascades begin with its molecular coupling to the insulin receptor (IR), which exhibits a complex structural conformation and consists of two extracellular α subunits and two β transmembrane portions [80]. The IR has intrinsic tyrosine kinase capacity, and the connection of insulin leads to the autophosphorylation of specific tyrosine residues present at the β subunits. Following this, several molecular messengers may be triggered by the IR, such as insulin-like growth factor (IGF-1), members of the insulin receptor substrate (IRS) family, *Shc* adapter protein isoforms, SIRP family peptides, in addition to Gab-1 (GRB2 associated binding protein 1), p60, Cbl (cannabinoid receptor 1), and APS (adapter protein with a PH and SH2 domain) proteins [81,82,83].

Insulin binding on the *β*-subunit of IR along with the consequent coactivation of IGF-1 and IRS1/2 at the tyrosine residues promote the activation of PI3K, as well as subsequent phosphorylation of protein kinase B (Akt) (reviewed in [84]). Akt activation leads to phosphorylation and nuclear exclusion of forkhead box O1 (FKHR or FoxO1), a key transcription factor that activates the expression of gluconeogenesis genes [85]. PI3K, phosphoinositide-dependent protein kinase-1 (PDK-1), and Akt in the serine/threonine-kinase-phosphorylated state, referring to molecular members of the first molecular pathway (I). When activated, PI3K results in Akt activation, thus activating the mTOR protein via the Rheb GTP binding protein (RHEB) (reviewed in [73,74,75,86]).

The energy-sensing signaling branch (II) is also a signaling pathway and regulates energy supply (Figure 2). When the cytoplasmic adenosine triphosphate (ATP) levels are reduced, or even insufficient, the serine/threonine protein kinase STK11 (also known as liver kinase B1, LKB1) triggers and phosphorylates the peptide AMPK (Activated Protein Kinase by Adenosine Monophosphate). As a result, the mTOR is inhibited by the TSC2 (Tuberous Sclerosis Complex 2). When ATP levels are restored or normalized to sustain cellular needs, mTOR inhibition is suppressed by the decreased activity of the LKB1 complex, as previously reported [73,74,75,86]. Furthermore, ketone body production from the liver is used by peripheral tissues as a source of energy during fasting [62]. Indeed, mTORC1 decreases its activity in response to fasting circumstances, thereby allowing greater PPAR-α activity (Peroxisome Proliferator-Activated Receptor Alpha), the main transcriptional regulator of ketogenic genes [87].

In turn, in the amino acid-sensitive signaling pathway (III), the Rags-Related GAP complex (GATOR1) regulates the mTOR expression in response to changes in amino acid levels, especially leucine and arginine. When levels of these amino acids are low, the GATOR1 protein complex inhibits mTOR signaling. Instead, when the leucine and arginine levels are high or normalized, mTOR inhibition by GATOR1 is then suppressed (reviewed in [73,74,75,86]).

Hyperactivation of the Akt/mTORC1 pathway induces lipogenesis by regulating the action of sterol regulatory element binding protein-1 (SREBP-1) in several steps [21,88]. SREBP-1 is a transcriptional regulator of insulin-stimulated fatty acid synthesis [21,88]. For example, the reduced expression of S6K1, an mTOR-activated substrate, has been shown to reduce SREBP-1 expression and hepatic triglyceride accumulation [76]. Therefore, SREBP1 expression is regulated by the mTOR-S6K axis [89]. Furthermore, SREBP-1 is a transcription factor that regulates the expression of genes responsible for lipid syntheses, such as acetyl-CoA carboxylase (ACC) and fatty acid synthase (FAS) (reviewed in [90]). In turn, ACC mediates the conversion of acetyl-CoA to malonyl-CoA, and FAS converts malonyl-CoA into palmitate, which can be esterified to triacylglycerols and stored in the liver tissue (reviewed in [91]).

## 4. Effects of mTOR Protein Signaling in the Liver in Response to Exercise Training Interventions

Exercise training prescription is complex, and health effects vary according to the variables used in terms of frequency, intensity, and duration parameters. Depending on the demands of each exercise training intervention, several adaptive effects and phenotypic changes might occur [1,92,93,94]. In this context, Table 1 presents information about the potential effects of exercise training protocols from studies with animal models.

Maintenance of AMPK and mTOR levels; unaffected hepatic triglyceride content

Bayod et al. (2014) [95] reported increased mTOR activation in the liver of the rats which exercised on a treadmill for 36 weeks (4–5 days a week). The exercised animals showed high levels of the p62/SQSTM1 protein, which may be related to metabolic effects, including the regulation of adipogenesis and energy balance [95]. After an intervention covering 32 weeks on a regular treadmill (60 min/day, 5 times/week), Piguet et al. (2015) [96] observed higher phosphorylated AMPK and its Raptor substrate, associated with lower mTOR activity. The AMPK activation increased immediately after exercise, followed by higher Ser^792^ phosphorylation on the Raptor complex, resulting in decreased mTORC1 activity, and consequent inhibition of lipogenesis. Indeed, the AMPK peptide acts in the following two ways to inhibit mTOR activity [96]: phosphorylation on the Thr^1227^ and Ser^345^ sites of TSC2, or inhibitory regulation on the Ser^722^ and Ser^792^ sites of the mTORC1 Raptor component. The TSC2 inhibitory mechanism is associated with the higher GTPase activity towards the Rheb protein (Ras homolog), resulting in an inactivation and a decrease in the mTORC1 signaling [103], following the scheme in Figure 2.

Furthermore, RNA sequencing in the hepatic samples from the animals immediately euthanized after an acute exercise training intervention revealed substantial changes in the following six molecular pathways involved with lipid metabolism: fatty acid triacylglycerol and ketone body metabolism, fatty acid β oxidation, PPAR (Peroxisome Proliferator-Activated Receptor) signaling pathway, regulation of the cellular ketone metabolic process, NAFLD (non-alcoholic fatty liver disease) pathway, and PI3K/Akt signaling [96]. Therefore, adaptive alterations in response to exercise training interventions may include multiple effects in terms of molecular activation/inhibition [1].

Rocha et al. (2017) [97] tested three overtraining protocols in mice and found that all exercise models resulted in higher fat concentration in the liver. In particular, the downhill-running-induced overtraining model increased the activation of key proteins from the mTOR signaling pathway. Such a response occurred despite the high levels of phosphorylated AMPK (Thr^172^) and SREBP-1 protein (p^125^) isoforms, which culminated in cellular edema associated with acute inflammation. According to the authors, fatty accumulation in the liver would be an impairment of insulin signal transduction in the skeletal muscle, promoted by exhaustive exercise. As a result of decreased glucose uptake in the skeletal muscles, glycemic metabolism would be redirected to the liver, where glucose could be converted to triacylglycerol, leading to fatty accumulation. Thus, any metabolic effect from exercise training interventions could be more pronounced when associated with greater workloads, resulting from increases in rest periods, as well as the number of sets within a session and/or frequency of workouts per week. A mismatch among different exercise training parameters could modulate the phenotypical response [1]. 

In contrast, Teglas et al. (2020) [98] found unaffected mTOR expression and reduced AMPK phosphorylation when subjecting mice to interval running exercise for 20 weeks. The exercise intervention was performed four times a week for 60 min, with each cycle consisting of 4 minutes at a high intensity (20 m/min) and 2 minutes at a low intensity (10 m/min), in a total of 10 cycles per training session. However, the authors could not establish a clear relationship between AMPK and mTOR expressions, although AMPK phosphorylation is associated with mTOR inhibition [85]. These results allowed us to hypothesize that exercise intervention did not reduce ATP levels to stimulate AMPK phosphorylation, consequently inhibiting the mTOR activation.

Tu et al. (2020) [8] investigated the effects of four weeks of treadmill exercise and the administration of rapamycin on energy metabolism gene expression in the hepatic tissue of rats previously supplemented with a high-fat diet (HFD). The authors found lower levels of phosphorylated S6K1, a marker of mTORC1 pathway activity, and reduced hepatic triglyceride content in response to the exercise protocol. According to the authors, although exercise reduced TG content and upregulated mitochondrial metabolic gene expression in the liver of HFD rats, this mechanism may not involve the mTOR pathway. Indeed, other important genes were not investigated, such as PPARγ (peroxisome proliferator-activated receptor gamma) and FABP4 (fatty acid binding protein 4). Moreover, Akt phosphorylation has been related to mTORC2 signaling, and it was not modified by the exercise protocol [8]. Guarino et al. (2020) [99] found higher phosphorylation of AMPK, followed by lower S6K1 expression and hepatic triglycerides levels in mice with non-alcoholic steatohepatitis. As mTOR expression was unaffected by the exercise intervention, the authors postulated that other pathways could be more directly associated with the hepatic adaptive alterations in response to exercise.

Similarly, Kwon et al. (2020) [100] found higher expression levels of phosphorylated AMPK in the liver in C57BL/6 male mice in response to a short protocol, consisting of five weekly sessions of endurance exercise by treadmill running (15 m/min). In contrast, the authors identified anabolic activation via higher phosphorylated Akt and mTOR levels in response to an exercise protocol. These results were followed by higher cellular autophagy and lower apoptosis. Importantly, the authors offered some remarks concerning divergent results from other studies; based on this, different intervals between the end of the exercise training and euthanasia could diversely affect molecular results.

Corroborating this postulation, Pinto et al. (2021) [101] investigated the acute effects of different physical exercise protocols, such as resistance, exhaustive, strength, and concurrent exercise training on autophagy markers, genes, and proteins. No significant acute change (immediately after training) was detected concerning the mTOR gene levels in the liver in response to the different protocols. In contrast, animals euthanized 6 h after the end of concurrent training presented lower levels of mTOR gene expression compared to strength training. Considering another study, Pinto et al. (2022) [103] identified no significant effects of a treadmill running protocol on the AMPK and mTOR expression results, as well as triglyceride hepatic content in C57BL/6 male mice.

Based on these findings, adaptive changes in the liver related to mTOR expression in response to exercise training interventions are still poorly understood. According to Watson and Baar [1], a potential mTOR signaling regulation is associated with reduced energy support and ATP levels. In response to the low ATP levels, the current mechanism consists of AMPK stimulation, which inhibits the mTOR and posterior lipogenesis. The other two mTOR signaling pathways are related to the growth factor pathway and amino acid sensor pathway, respectively [73]. Depending on cross-talks among these three known mTOR molecular pathways, the tissue phenotype may differ. Importantly, molecular interactions are regulated not only by nutritional availability and/or energy substrate but also by different exercise training variables, such as type, intensity, duration per session, and frequency. It is noteworthy that previous studies have used several exercise protocols, as well as diverse animal models, which could sustain potential discrepancies among investigations. Generally, mTOR signaling seems to be more sensitive to changes in response to specific conditions, including high volume [95], exhaustive/very intense, and shorter exercise training [97,100], or upon concurrent intervention [101] (Figure 3).

## 5. Future Directions

Further studies should investigate the potential effects of different exercise training categories, including resistance, endurance, and concurrent exercise protocols, as well as high-intensity interval training, on the mTOR signaling pathways and phenotypical adaptive changes in the liver in different experimental models.

## 6. Conclusions

Exercise training interventions have been associated with multiple metabolic and tissue benefits. However, it is worth highlighting that the mTOR signaling in response to exercise training in the liver remains unclear. Indeed, hepatic adaptive alterations seem to be most outstanding when sustained by chronic/long interventions of moderate intensity, or short high-intensity exercise protocols.

## Figures and Tables

**Figure 1 biology-13-00362-f001:**
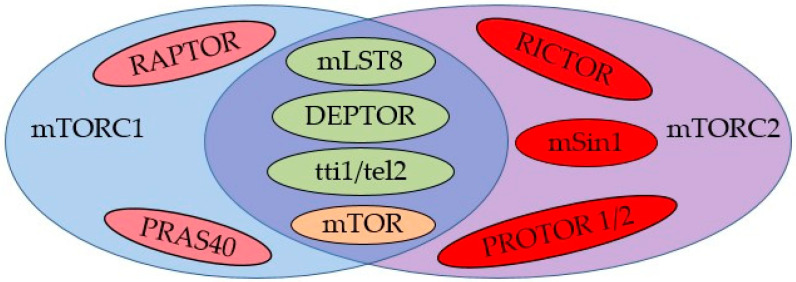
Schematic presentation of the mTORC1 and mTORC2 structures. Both peptides have four common protein structures: mLST8, DEPTOR, tti1/tel2, and mTOR. RAPTOR, regulatory-associated protein of mammalian target of rapamycin; PRAS40, proline-rich Akt substrate of 40 kDa; DEPTOR, serine/threonine kinase mTOR DEP domain-containing mTOR-interacting protein; mLST8, mammalian lethal with SEC13 protein 8; tti1/tel2 peptide; mTOR, mammalian target of rapamycin; RICTOR, rapamycin-insensitive companion of mTOR; mSin1, mammalian stress-activated map kinase-interacting protein 1; protein observed with RICTOR 1 and 2 (PROTOR 1/2). Illustration prepared and adapted from previous studies [25,26,37]; additional details may be found in Laplante and Sabatini (2012) [25], Saxton and Sabatini (2017) [26], and Liu and Sabatini (2020) [37].

**Figure 2 biology-13-00362-f002:**
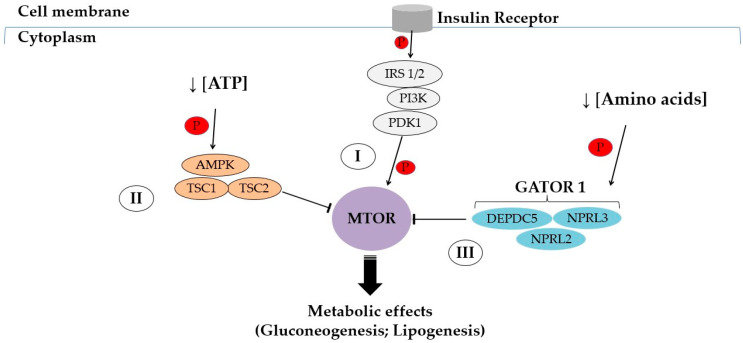
Schematic presentation of mTOR modulation pathways. Three primary signaling pathways are known to converge to modulate the action of the mTOR peptide. Depending on metabolic, hormonal, and nutritional conditions, crosstalk mechanisms supported by these pathways regulate mTOR stimulation and activity. (I) Growth factor signaling pathway: IRS 1/2, subtypes 1/2 of Insulin Receptor Substrate; PI3K, Phosphoinositide-3-Kinase; PDK1, Phosphoinositide-Dependent Protein Kinase-1; (II) ATP/energy sensitive signaling pathway: AMPK, Activated Protein Kinase by Adenosine Monophosphate; TSC1, Tuberous Sclerosis Complex-1; TSC2, Tuberous Sclerosis Complex-2; and (III) Amino Acid-Sensitive Signaling Pathway: GATOR 1, Rags-Related GAP Complex; DEPDC5, DEP protein, containing domain 5; NPRL2, Nitrogen Permease Regulatory protein type 2; NPRL3, Nitrogen Permease Regulatory protein type 3. Illustration adapted from previous studies [73,74,75]; for more complete details, see Han and Wang (2018) [73], Ma and Blenis (2009) [74], and Crino (2016) [75].

**Figure 3 biology-13-00362-f003:**
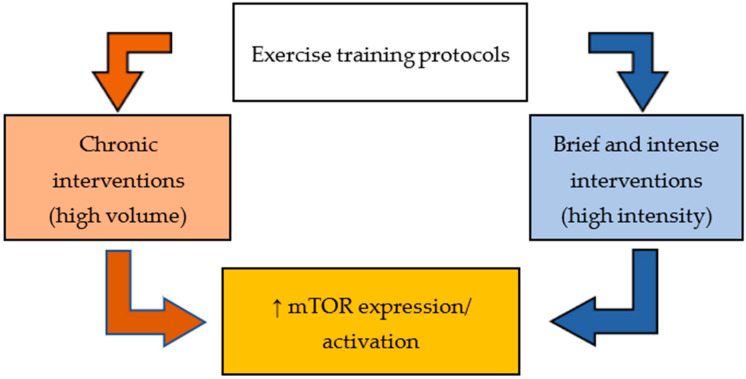
Differential effects of exercise training on mTOR expression and/or activation in liver; ↑, increase.

**Table 1 biology-13-00362-t001:** Effects of exercise training protocols on mTOR signaling in the liver in animal models.

Authors	Exercise Protocol	Animal	Outcomes
Bayod et al. (2014) [95]	Treadmill, 36 weeks, 4–5 days a week for 30 min. Intensity: moderate without increments, speed 12 m/min.	Sprague–Dawley rats (5 weeks old)	↑ mTOR and p62/SQSTM1 activation in the exercised group
Piguet et al. (2015) [96]	Treadmill, a single bout of exercise for 60 min. Intensity: low to moderate, no increments, vel. 12.5 m/min. Animals were euthanized 15 min after the end of the exercise training.	Hepatocyte-specific PTEN-deficient mice (*AlbCrePten ^phlox/phlox^*) (7 weeks old)	↑ AMPK phosphorylation ↓ S6K1 phosphorylation ↓ mTOR activity
Rocha et al. (2017) [97]	Treadmill, 8 weeks, 5 times a week. Intensity: high, with load increments after 4 weeks. Animals were euthanized 36 h after the end of the exercise training.	C57BL/6 male mice (8 weeks old)	↑ activation of Akt, mTOR, and S6K1 ↑ percentage of fat
Téglás et al. (2020) [98]	Treadmill (interval running), 20 weeks, 4 times a week for 60 min. Intensity: low (10 m/min) and high (20 m/min), without increments.	Male APP/PS1 transgenic mice (12 weeks old)	Unaffected mTOR expression ↓ AMPK phosphorylation
Tu et al. (2020) [8]	Treadmill, 4 weeks, 5 times a week for 20–40 min. Intensity: high.	Sprague-Dawley rats (5–6 weeks old) submitted to a high-fat diet	↓ activation of S6K1 ↓triglycerides content, and unaffected Akt activation
Guarino et al. (2020) [99]	Treadmill, 8 weeks, 5 times a week for 60 min. Intensity: low–moderate (12.5 m/min). Animals were euthanized 48 h after the end of the exercise training.	C57BL/6 male mice (8 weeks old) with non-alcoholic steatohepatitis (NASH)	↑ AMPK Unaffected mTOR expression; ↓ S6K1 ↓ hepatic triglyceride content
Kwon et al. (2020) [100]	Treadmill, 5 days, 1 time per day for 60 min. Intensity: 15 m/min. Animals were euthanized 90 min after the end of the exercise training.	C57BL/6 male mice (9 weeks old)	↑ mTOR phosphorylation ↑ AMPK phosphorylation ↑ ULK phosphorylation (autophagy)
Pinto et al. (2021) [101]	A single bout of exercise, treadmill (60% of exhaustion speed) and climbing (75% of body weight). Intensity: endurance (60 min at 0% incline), high (until exhaustion at 14% incline), strength training (10 climbs), and concurrent (5 climbs and 30 min run). Euthanasia was carried out at the following two points: immediately (acute), and 6 h after the end of training.	C57BL/6 male mice (6 weeks old)	Maintenance of mTOR levels as an acute response to exercise (animals euthanized immediately after training); ↓ mTOR expression associated with concurrent training concerning strength training; indeed, a higher interval period (6 h) was associated with greater mTOR expression than the acute period.
Pinto et al. (2022) [102]	Treadmill (endurance or exhaustive), 8 weeks, 5 times a week, progressive intensity (60% to 75%). Animals were euthanized 36 h after the end of the exercise training.	C57BL/6J male mice (6 weeks old)	Maintenance of AMPK and mTOR levels; unaffected hepatic triglyceride content

↑, increase; ↓, reduce.

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
