# Peer review of "Exercise, mTOR Activation, and Potential Impacts on the Liver in Rodents"

_biology, 2024, doi:10.3390/biology13060362_

Round 1
Reviewer 1 Report (New Reviewer)
Comments and Suggestions for Authors
In attachment

Author Response
Reviewer #1
Exercise, mTOR activation and potential impacts on the liver
The authors describe basic information about the structure and function of mTOR and the signalling pathways that could influence liver metabolism. The final part summarizes studies in animal models showing the effects of exercise on mTOR activation in the liver. The review is generally well conceived and provides general information on the issues under study, but I see several major formal weaknesses.
Author's answer. Thank you for carefully reviewing our manuscript and for helpful suggestions.
Major points:
1) The title should state that there are exclusively studies on animal models.
Author's answer. Thanks for scoring that. The title was adjusted.
2) The last sentence in the abstract should be corrected because the whole of section 5 (“Effects of mTOR protein signaling in the liver in response to ET interventions”) shows that the effects of exercise on mTOR and lipogenesis in the liver are inconsistent, and it is not at all clear that there is an improvement in this pathway, which the authors themselves admit in the conclusion.
Author's answer. We thank the reviewer for pointing this out. This point was reviewed in abstract.
3) In the introduction, lines 35-39 refer to "human health", but reference (1) refers to a study on rats. It would be useful to give a reference that is on humans or to change the wording. Similarly, line 46-47 - the claim again appears to be human data, but the references cite rat/mouse studies. Please change the wording to make this obvious, or cite human studies.
Author's answer. We are very grateful for this consideration, and citations were reviewed and corrected.
4) Chapters 2-3 contain mostly citations of reviews, specifically citations 18, 19, 20, 21, 22, 27, 28, 35, 36, 44, 78 etc. I think this is not very appropriate and it would be better to directly cite the source/original works - where possible, or to state that this issue “is reviewed in…”
Author's answer. Thanks for scoring that. Reviews were replaced by original citations. When this adjustment was not possible, we optioned maintaining reviewing studies. Therefore, the References section was reviewed and adapted in accordance with the current appointment.
5) Importantly, there are several statements in the text that do not correspond to the citations: e.g., line 131 - citation 43 is about hepatic mTORC2 signalling but not about "brown adipose tissue"; line 132 - citation 44 is about IGF-1 signalling in muscle, but there is no clear link to mTORC2 deficiency in liver and IR; furthermore, the text seems not to correspond to citation 65 (line 205-205), citation 25, 26 (line 79-80), etc. Since it was not possible for me to check all the references, I recommend checking ALL the citations and editing appropriately.
Author's answer. Thanks for scoring that. All citations were reviewed and edited according to the situation. As a result, the bibliography sequence was changed.
6) Several sentences/statements are not understandable or not correctly worded/interpreted, specifically line 165-167, 179, 185, 186, 213.
Author's answer. We made alterations in textual presentation.
Minor points:
1) The abstract contains the wording "..several adaptive alterations in multiple body tissues, such as the liver..", which is not precise. The article focuses only on the liver, so it should be, in my opinion, "several adaptive alterations in the liver" and again "in animal models”.
Author's answer. We made alterations in textual presentation. Thank you.
2) The numbering of sections (parts) should be corrected - section 3 is followed by section 5.
Author's answer. Thanks for scoring that.
3) It is not very clear to me why this sentence on the specific regulation of IR occurs in the introduction: “Relative to the liver, ET intervention attenuated obesity-induced insulin 41 resistance, by inhibitory regulation of cardiolipin synthase 1 (CRLS1)/ 42 interferon-regulatory factor-2 binding protein 2 (IRF2bp2)-activating transcription factor 43 3 (ATF3) pathway cascade [3]. “ The effects of ET on insulin resistance are much more complex, and certainly not limited to this pathway.
Author's answer. Thanks for scoring that. The original text was reviewed and changed.
4) The font size of Ttil1/TTel2 in the text and in the legend of figure 1 is different. Please check and unify.
Author's answer. Thanks for scoring that.
5- On line 178 there is a reference to the figure marked "(II)", which is not clear, there should be a reference to "Fig 2 (II)".
Author's answer. We made alterations in textual presentation. Thank you.
Reviewer 2 Report (New Reviewer)
Comments and Suggestions for Authors
The manuscript entitled "Exercise, mTOR activation, and potential impacts on the liver" addresses a topic of considerable interest to the scientific community, namely the complex interplay between exercise-induced mTOR activation and its potential consequences for liver physiology, particularly in the context of fatty liver disease. Exploring the multifaceted relationship between exercise, mTOR activation and hepatic homeostasis is of great importance for understanding metabolic regulation and potential therapeutic interventions.
Although limited by the paucity of recent investigations in this area, the authors have carefully curated and analysed the available literature, providing valuable insights into the current scientific discourse surrounding the link between exercise-induced mTOR activation and liver physiology. However, the introductory sections of the manuscript, while providing essential contextualisation of signalling cascades, could benefit from brevity to optimise the discourse space for a more exhaustive exploration of the primary research focus.
Furthermore, the section devoted to the core thematic investigation occupies a modest proportion of the manuscript. A greater emphasis on this area of focus would enhance the scientific rigour and depth of the manuscript, making it more in line with scholarly expectations.
In conclusion, "Exercise, mTOR activation, and potential impacts on the liver" is a good scientific paper that elucidates the complex interplay between exercise-induced mTOR activation and hepatic physiology.
Given the complexity and variability that characterise research findings in this area, the inclusion of graphical representations in the conclusion section of the manuscript could serve to enhance data visualisation and elucidate key findings, thereby increasing the scientific accessibility and impact of the manuscript.
Author Response
Reviewer #2
The manuscript entitled "Exercise, mTOR activation, and potential impacts on the liver" addresses a topic of considerable interest to the scientific community, namely the complex interplay between exercise-induced mTOR activation and its potential consequences for liver physiology, particularly in the context of fatty liver disease. Exploring the multifaceted relationship between exercise, mTOR activation and hepatic homeostasis is of great importance for understanding metabolic regulation and potential therapeutic interventions.
Author's answer. Thank you for carefully reviewing our manuscript and for helpful suggestions.
1) Although limited by the paucity of recent investigations in this area, the authors have carefully curated and analysed the available literature, providing valuable insights into the current scientific discourse surrounding the link between exercise-induced mTOR activation and liver physiology. However, the introductory sections of the manuscript, while providing essential contextualisation of signalling cascades, could benefit from brevity to optimise the discourse space for a more exhaustive exploration of the primary research focus. Furthermore, the section devoted to the core thematic investigation occupies a modest proportion of the manuscript. A greater emphasis on this area of focus would enhance the scientific rigour and depth of the manuscript, making it more in line with scholarly expectations.
Author's answer. We thank the reviewer for pointing this out. The Introduction section was reviewed, and additional information was inserted in the new version.
2) In conclusion, "Exercise, mTOR activation, and potential impacts on the liver" is a good scientific paper that elucidates the complex interplay between exercise-induced mTOR activation and hepatic physiology.
Given the complexity and variability that characterise research findings in this area, the inclusion of graphical representations in the conclusion section of the manuscript could serve to enhance data visualisation and elucidate key findings, thereby increasing the scientific accessibility and impact of the manuscript.
Author's answer. A novel picture was inserted in the new textual version. Thank you.
Round 2
Reviewer 1 Report (New Reviewer)
Comments and Suggestions for Authors
The authors have taken into account all the comments and have greatly improved the citations in the manuscript and the overall presentation.
I would probably suggest a revision of the English language and a formal correction of the text (e.g. in line 36 - the "traning exercise" is repeated twice, etc.), but after this minor revision I recommend the manuscript for publication.
Author Response
Reviewer #2
The authors have taken into account all the comments and have greatly improved the citations in the manuscript and the overall presentation.
I would probably suggest a revision of the English language and a formal correction of the text (e.g. in line 36 - the "training exercise" is repeated twice, etc.), but after this minor revision I recommend the manuscript for publication.
Author's answer. Thank you for carefully reviewing our manuscript and for helpful suggestions. The text was reviewed, and English language was corrected by a Linguistic Reviewer (attached file).

This manuscript is a resubmission of an earlier submission. The following is a list of the peer review reports and author responses from that submission.
Round 1
Reviewer 1 Report
Comments and Suggestions for Authors
Reviewer 2 Report
Comments and Suggestions for Authors
The manuscript entitled "Exercise, mTOR activation and potential impacts on the liver". Title, abstract and overall rationale of work to some extent is good and explain detail about the mTOR role in liver. However, there are still some major concerns, which needs to be addressed.
1) Keywords: Author must be add some more keywords in the revised version of the manuscript
2) Introduction section is very short and author must be elaborate this section
3) In this section ( MTOR complex) author must add more section and write details about the role of these six protein component in MTOR and role in liver
Serine/threonine kinase MTOR,
MTOR regulatory protein (Raptor),
Proline-rich AKT substrate of 40kDa (PRAS40),
DEP domain-containing MTOR-interacting protein (DEPTOR),
Mammalian lethal with SEC13 protein 8 (mLST8),
Tti1/tel2
4) In this section (MTORC1) author must incorporate mechanism figure and show the genes involvement and how they regulates the mechanism.
5) In this section (Effects of mTOR protein signaling in the liver in response to exercise training) author add the figure to show clear pictorial view of this all paragraph (Mechanism). I saw the table there is mention but it is not impressive and attractive.
6 Conclusion section author must be elaborate this section should present at least in one 250-300 words. Author also need to write future prospective of this study.
7) There are lot of punctuation and typographical errors throughout in the manuscript. Please correct it.
8) Some references are too old author must be replace and add new for example
Carvalheira, J.B.C.; Zecchin, H.G.; Saad, M.J.A. Vias de Sinalização Da Insulina. Arch Endocrinol Metab 2002, 46, 419–425, 411
47. Alessi, D.R.; James, S.R.; Downes, C.P.; Holmes, A.B.; Gaffney, P.R.J.; Reese, C.B.; Cohen, P. Characterization of a 4133-Phosphoinositide-Dependent Protein Kinase Which Phosphorylates and Activates Protein Kinase Bα. Curr Biol 1997, 7, 261– 414 269,